# Evaluation of Advanced Artificial Intelligence Algorithms’ Diagnostic Efficacy in Acute Ischemic Stroke: A Comparative Analysis of ChatGPT-4o and Claude 3.5 Sonnet Models

**DOI:** 10.3390/jcm14020571

**Published:** 2025-01-17

**Authors:** Mustafa Koyun, Ismail Taskent

**Affiliations:** 1Department of Radiology, Kastamonu Training and Research Hospital, Kastamonu 37150, Turkey; 2Department of Radiology, Kastamonu University, Kastamonu 37150, Turkey; dr.taskent@gmail.com

**Keywords:** ChatGPT, Claude, artificial intelligence, acute ischemic stroke, magnetic resonance imaging, radiology

## Abstract

**Background/Objectives:** Acute ischemic stroke (AIS) is a leading cause of mortality and disability worldwide, with early and accurate diagnosis being critical for timely intervention and improved patient outcomes. This retrospective study aimed to assess the diagnostic performance of two advanced artificial intelligence (AI) models, Chat Generative Pre-trained Transformer (ChatGPT-4o) and Claude 3.5 Sonnet, in identifying AIS from diffusion-weighted imaging (DWI). **Methods:** The DWI images of a total of 110 cases (AIS group: *n* = 55, healthy controls: *n* = 55) were provided to the AI models via standardized prompts. The models’ responses were compared to radiologists’ gold-standard evaluations, and performance metrics such as sensitivity, specificity, and diagnostic accuracy were calculated. **Results:** Both models exhibited a high sensitivity for AIS detection (ChatGPT-4o: 100%, Claude 3.5 Sonnet: 94.5%). However, ChatGPT-4o demonstrated a significantly lower specificity (3.6%) compared to Claude 3.5 Sonnet (74.5%). The agreement with radiologists was poor for ChatGPT-4o (κ = 0.036; %95 CI: −0.013, 0.085) but good for Claude 3.5 Sonnet (κ = 0.691; %95 CI: 0.558, 0.824). In terms of the AIS hemispheric localization accuracy, Claude 3.5 Sonnet (67.2%) outperformed ChatGPT-4o (32.7%). Similarly, for specific AIS localization, Claude 3.5 Sonnet (30.9%) showed greater accuracy than ChatGPT-4o (7.3%), with these differences being statistically significant (*p* < 0.05). **Conclusions:** This study highlights the superior diagnostic performance of Claude 3.5 Sonnet compared to ChatGPT-4o in identifying AIS from DWI. Despite its advantages, both models demonstrated notable limitations in accuracy, emphasizing the need for further development before achieving full clinical applicability. These findings underline the potential of AI tools in radiological diagnostics while acknowledging their current limitations.

## 1. Introduction

Stroke is defined by the American Heart Association/American Stroke Association (AHA/ASA) as a neurological deficit secondary to the acute focal injury of the central nervous system (CNS) caused by conditions such as cerebral infarction, intracerebral hemorrhage (ICH), and subarachnoid hemorrhage (SAH) [1]. It is reported that stroke is more common in elderly individuals, with the average age being 69 years [2]. Acute ischemic stroke (AIS) accounts for 65% of all stroke cases worldwide, with this proportion reaching up to 87% in some studies [3,4]. AIS, a significant cause of disability and mortality worldwide, affects approximately 57.93 million people annually [5]. However, AIS is a treatable condition when addressed in its early stages [6]. Patients in the post-stroke period typically experience physical movement limitations, communication disorders, and cognitive function losses [6]. These multiple impairments cause patients to require comprehensive rehabilitation and intensive medical care. When AIS is intervened in the early period, it is a treatable condition, and, therefore, its early detection is critically important [7]. AIS can be treated with a medical approach called “thrombolysis” using alteplase (also known as tissue Plasminogen Activator (tPA)) or Tenecteplase (TNK) [2]. Another treatment option for AIS is mechanical thrombectomy, and studies are being conducted aiming to reduce the door-to-puncture time (DTPT) to enable rapid intervention [8].

While various radiological imaging techniques can be used to detect AIS, diffusion-weighted magnetic resonance imaging (DW-MRI) is the most accurate method [9]. However, recent studies suggest that non-contrast computed tomography (NCCT) and computed tomography angiography (CTA), which can be performed more quickly than DW-MRI, should be primarily implemented to detect AIS and rule out intracranial hemorrhage (ICH) so that treatment can be initiated without delay [2,8].

Artificial intelligence (AI) is a sophisticated branch of technology that mimics human intelligence and has brought about revolutionary changes across various fields. This technology can generate original content in multiple forms, including text, images, music, and video. Applicable in nearly every sector—from healthcare to finance, education to transportation—AI stands out for its ability to solve complex problems, perform data analysis, and make predictions. The use of AI in healthcare covers a broad area [10]. CardioAI, which analyzes cardiac magnetic resonance images, and computer-aided diagnosis (CAD) systems used in endoscopy–colonoscopy procedures [11] can be given as examples of AI applications in healthcare.

AI technology is divided into various specialized subdisciplines, including machine learning (ML), deep learning (DL), and computer vision [11]. ML identifies patterns using specific features and optimizes decision-making processes by applying this knowledge to similar situations. This technology has evolved into DL, which utilizes artificial neural networks (ANNs) mimicking human neural systems. Image processing technology encompasses the digital perception and analysis of visual data [11]. Two of the most prominent examples of AI are Chat Generative Pre-trained Transformer (ChatGPT) (OpenAI, San Francisco, CA, USA) and Claude (Anthropic, San Francisco, CA, USA), which have made groundbreaking advancements in the field of natural language processing. ChatGPT, developed by OpenAI and introduced on 30 November 2022, is one of the large language models (LLMs) capable of generating human-like text, responding to complex questions, and leveraging an extensive knowledge base [12]. On the other hand, Claude, launched by Anthropic in March 2023, is an LLM designed with a particular emphasis on ethics and safety, incorporating a more controlled and transparent approach [13]. Both models excel in interacting naturally with users, providing information on a wide range of topics, and generating creative content. ChatGPT-4o (ChatGPT-4 Omni) and Claude 3.5 Sonnet models have been recently developed and offer the ability to evaluate visual materials.

The integration of AI technology into radiological imaging systems holds significant potential for accelerating disease diagnosis processes, enhancing workflow efficiency, and minimizing errors caused by human factors [14]. Recent studies demonstrate the effectiveness of DL applications in detecting various pathologies from radiological imaging data [15,16,17]. DL applications are also coming to the forefront in AIS detection (Table 1) [18,19,20,21]. It is reported that sensitivity and specificity values above 80% have been achieved with these AI applications. However, there are only a few studies on the detection of pathologies from radiological images using ChatGPT [22,23,24,25], and, among these, only the study by Kuzan et al. focused on AIS detection from DW-MRI images [25]. However, to the best of our knowledge, no studies have been conducted on the detection of pathologies from radiological images using Claude.

The aim of this study is to investigate the detectability of AIS from DW-MRI images using the two most popular AI models, ChatGPT-4o and Claude 3.5 Sonnet, and to compare their diagnostic performance.

## 2. Materials and Methods

### 2.1. Selection of Cases

A total of 1256 patients with DW-MRI images recorded in our hospital’s information management system between May 2024 and September 2024 were retrospectively analyzed. These patients underwent systematic evaluation. Accordingly, patients over 18 years of age who presented to the emergency department with neurological symptoms suggestive of AIS (hemiparesis, aphasia, facial asymmetry, diplopia, and vision loss) and underwent DW-MRI on the same day with preliminary diagnosis of AIS were identified. Among the screened patients, 55 cases with diffusion restriction on DW-MRI were classified as the AIS group (Figure 1). Cases with clinical findings but no diffusion restriction on DW-MRI were not included in the AIS group (Figure 1).

The control group consisted of 55 healthy cases who underwent DW-MRI for various indications but showed no diffusion restriction (Figure 1).

The exclusion criteria for both groups were standardized as follows: the presence of artifacts limiting image evaluation, age under 18 years, a history of intracranial mass or recent cranial surgery, intracranial hemorrhage, or the presence of metallic objects (Figure 1). Cases were determined by consensus between two radiologists with 8 and 12 years of experience.

### 2.2. Brain DWI and ADC İmages

The brain DWI and ADC images for all cases were obtained using a 1.5T MRI scanner (GE Signa Explorer; GE Healthcare, Milwaukee, WI, USA).

### 2.3. AI Models Used in the Study

In our study, we used ChatGPT-4o (GPT-4 Omni), developed by OpenAI (OpenAI, San Francisco, CA, USA) and introduced on 13 May 2024, and Claude 3.5 Sonnet model, developed by Anthropic (Anthropic, San Francisco, CA, USA) and introduced on 20 June 2024, to analyze DW-MRI images.

### 2.4. Prompt Prepared for AI Models

A structured three-question prompt was developed for the images presented to the ChatGPT-4 and Claude 3.5 Sonnet models as follows:In these DWI and ADC images, is there an acute ischemic stroke? Please answer with “Yes” or “No”.If there is an acute ischemic stroke, what is the localization according to the patient? Please answer with “Right cerebral hemisphere”, “Left cerebral hemisphere”, “Right cerebellar hemisphere”, or “Left cerebellar hemisphere”.If there is acute ischemia, could you specify the affected specific lobe or region?

### 2.5. Image Selection, and Preparation and Presentation to Models

Image slice selection criteria for AIS cases were standardized with a methodological approach. DWI sequence slices showing the ischemic lesion at its maximum size and optimal visibility, along with their corresponding ADC maps, were designated as reference images for primary analysis. Parameters such as similar distribution numbers of anatomical lesions or similar types of clinical presentations were disregarded in image selection.

The selection of image slices for the healthy control group was systematically performed to show anatomical correlation with the reference slices in the AIS group. The spatial localization of image slices was standardized to match the slice levels in AIS cases as closely as possible. This methodological approach aimed to ensure that AI models analyzed equivalent anatomical structures and localizations in both groups.

The original Digital Imaging and Communications in Medicine (DICOM) format DWI and ADC images of the cases were first anonymized using AW Volumeshare 7 software (AW version 4.7, GE Healthcare, Milwaukee, WI, USA). Subsequently, the images were converted to JPEG format through this software without altering basic parameters such as resolution, brightness, and contrast. All images were exported while preserving the original pixel size. During conversion, high-level JPEG quality was preferred to preserve diagnostically important details as much as possible. Later, DWI and ADC images were combined into a single image using Microsoft Paint software (Microsoft, Corp., Redmond, WA, USA), and irrelevant regions outside the brain were cropped from the newly created image. Additionally, orientation verification was performed by radiologists on the images before uploading them to the models. To minimize bias, the cases were presented to the models in a randomized order, accompanied by the three-question prompt mentioned earlier. The responses provided by the models were recorded (Figure 2). Additionally, to assess intra-model reliability, all cases were reanalyzed with the same models two weeks later.

### 2.6. Evaluation of Outputs

To ensure standardization during the evaluation process, preliminary assessments were conducted on sample cases not included in the study to establish consensus on evaluation criteria among the radiologists. The response categorization criteria were defined as follows: a “correct” response required full agreement between the model’s answers and the radiologists’ reports, while a “wrong” response was defined as a mismatch in key findings or the presence of diagnostically critical errors.

Throughout the evaluation process, radiologists conducted their work using medical monitors on diagnostic imaging workstations.

### 2.7. Statistical Analysis

Statistical analyses were conducted using IBM SPSS 23 software (IBM Corporation, Armonk, NY, USA). The normality of data distribution was assessed with the Kolmogorov–Smirnov test. An Independent-Samples *t*-test was used to compare age between groups, while a Chi-square test was applied for gender distribution comparisons. Quantitative data with normal distribution were presented as mean and standard deviation (Mean ± SD), and categorical data were reported as frequency (n) and percentage (%).

The imaging analysis performance of the models was evaluated in comparison to radiologist assessments (gold standard). The performance of ChatGPT-4o and Claude 3.5 Sonnet was assessed by calculating sensitivity, specificity, positive predictive value (PPV), negative predictive value (NPV), and diagnostic accuracy rates. The agreement between the models and radiologists, as well as inter- and intra-model reliability, was assessed using Cohen’s kappa coefficient.

The interpretation of the kappa coefficient was as follows: ≤0.20 as “slight agreement”, 0.21–0.40 as “fair agreement”, 0.41–0.60 as “moderate agreement”, 0.61–0.80 as “substantial agreement”, and 0.81–1.00 as “almost perfect agreement”. The effect of the age variable on the models’ AIS detection performance was evaluated using binary logistic regression analysis, while the effect of the gender variable on model performance was assessed using the Pearson Chi-square test. A binomial test was used to compare the models’ predictions with random guessing. Statistical significance was set at *p* < 0.05.

### 2.8. Ethical Approval

This retrospective study was conducted in accordance with ethical guidelines and was approved by the Clinical Research Ethics Committee of Kastamonu University on 10 December 2024, under approval number 2024-KAEK-134. The study adhered to the Checklist for Artificial Intelligence in Medical Imaging (CLAIM) guidelines [26]. Due to the retrospective nature of the study, written informed consent was not obtained from the patients.

## 3. Results

### 3.1. Results Obtained from Demographic Data

A total of 110 cases were included in the study, with 55 cases in the AIS group and 55 in the healthy control group. Both groups consisted of 25 females (45.5%) and 30 males (54.5%). The mean age of the AIS group was 73.11 ± 11.43 years, while the mean age of the healthy control group was 72.75 ± 10.75 years. No significant differences were found between the AIS and healthy control groups in terms of age or gender (*p* > 0.05) (Table 2).

### 3.2. Diagnostic Performance of the Models

In the diagnostic performance analysis of ChatGPT-4o for AIS detection, all cases in the AIS group (100%) yielded true positive results. However, in the control group, only two cases (3.6%) produced true negative results, while 53 cases (96.4%) resulted in false positive outcomes (Table 3).

For Claude 3.5 Sonnet, the diagnostic performance for AIS detection showed 52 cases (94.5%) with true positive results and 3 cases (5.5%) with false negative results in the AIS group. In the control group, 41 cases (74.5%) produced true negative results, while 14 cases (25.5%) resulted in false positive outcomes (Table 3).

Both models demonstrated a high sensitivity for AIS detection (ChatGPT-4o: 100%, Claude 3.5 Sonnet: 94.5%), but specificity values differed significantly (ChatGPT-4o: 3.6%, Claude 3.5 Sonnet: 74.5%). The diagnostic accuracy of Claude 3.5 Sonnet (84.5%) was higher compared to ChatGPT-4o (51.8%). The binomial test revealed that ChatGPT-4o’s accuracy in detecting AIS (51.8%) was not statistically different from random guessing (50%) (*p* = 0.775). In contrast, the same test demonstrated that Claude 3.5 Sonnet’s accuracy in detecting AIS was 84.5%, which is significantly higher than the random guessing rate of 50% (*p* = 0.000). The positive predictive value (PPV), negative predictive value (NPV), and F1 score for both models are presented in Table 3.

Both models answered all the questions in the prompt consistently with each other. When the models responded that AIS was present, they answered the other questions accordingly, and, when they responded that AIS was not present, they left the other questions unanswered. The number of cases where ChatGPT-4o answered all questions with complete accuracy was 4 (7.3%), while, for Claude 3.5 Sonnet, it was 17 (30.9%) (Table 4).

The AIS detection performance of both models does not show any statistically significant differences between genders (*p* > 0.05). Likewise, no statistically significant differences were found between age and the models’ AIS detection performance (*p* > 0.05).

### 3.3. Agreement Analysis Between Models and Radiologists

ChatGPT-4o demonstrated slight agreement with the radiologists’ gold-standard assessment (κ = 0.036; %95 CI: −0.013, 0.085). In contrast, Claude 3.5 Sonnet showed substantial agreement with the radiologists (κ = 0.691; %95 CI: 0.558, 0.824) (Table 5).

### 3.4. Performance of the Models in Detecting Hemispheric Localization of AIS

ChatGPT-4o correctly identified the hemisphere of AIS localization in 18 out of 55 cases (32.7%) in the AIS group (16 in the right cerebral hemisphere and 2 in the left cerebral hemisphere). However, in the healthy control group, it incorrectly made localization predictions in 53 out of 55 cases (96.4%) despite the absence of AIS (Table 6).

In terms of hemispheric localization, ChatGPT-4o demonstrated slight agreement with radiologists (κ = 0.077; %95 CI: −0.151, −0.003).

Claude 3.5 Sonnet correctly identified the hemisphere of AIS localization in 37 out of 55 cases (67.3%) in the AIS group (17 in the right cerebral hemisphere and 20 in the left cerebral hemisphere). However, in the healthy control group, it incorrectly made localization predictions in 14 out of 55 cases (25.5%) despite the absence of AIS (Table 7).

In detecting AIS localization by hemisphere, Claude 3.5 Sonnet demonstrated moderate agreement with the gold-standard radiologists (κ = 0.572; %95 CI: 0.456, 0.688). Additionally, Claude 3.5 Sonnet was statistically significantly more successful than ChatGPT-4o in accurately identifying AIS localization by hemisphere (*p* = 0.0006).

### 3.5. Performance of the Models in Detecting Specific AIS Localization

ChatGPT-4o correctly identified the specific localization of AIS in only 4 cases (7.3%) within the AIS group, while making incorrect determinations in the remaining 51 cases (92.7%). In contrast, Claude 3.5 Sonnet accurately identified the specific localization in 17 cases (30.9%) within the AIS group, but made incorrect determinations in 38 cases (69%) (Table 8).

In detecting the specific lobe or region of AIS localization, ChatGPT-4o demonstrated slight agreement with radiologists (κ = 0.026; %95 CI: −0.015, 0.067), while Claude 3.5 Sonnet showed fair agreement (κ = 0.365; %95 CI: 0.277, 0.453). Claude 3.5 Sonnet was found to be statistically significantly more successful than ChatGPT-4o in accurately identifying the specific lobe or region of AIS localization (*p* = 0.0016).

### 3.6. Second Round of Model Evaluations and Intra-Model Agreement Analysis

After a two-week interval, the DWI-ADC images were re-evaluated by the models. In the second evaluation, ChatGPT-4o again showed 100% true positive results for all AIS cases. In the healthy control group, 48 cases (87.3%) resulted in false positives, while 7 cases (12.7%) produced true negative results (Table 9). Claude 3.5 Sonnet correctly identified 46 cases (83.6%) as true positive and 9 cases (16.4%) as false negative in the AIS group. In the healthy control group, 46 cases (83.6%) were identified as true negative, and 9 cases (16.4%) were false positive (Table 7).

In the second evaluation, ChatGPT-4o showed an increase in specificity, positive predictive value (PPV), and diagnostic accuracy (Table 9). On the other hand, Claude 3.5 Sonnet showed a decrease in sensitivity, negative predictive value (NPV), and diagnostic accuracy, while the specificity and PPV values increased (Table 9).

Moderate agreement was found between the two evaluations for both ChatGPT-4o and Claude 3.5 Sonnet in detecting the presence of AIS (κ = 0.428; %95 CI: 0.026, 0.830 for ChatGPT-4o, κ = 0.582; %95 CI: 0.433, 0.731 for Claude 3.5 Sonnet).

In the assessment of the hemispheric localization of AIS, fair agreement was found between the first and second round results of ChatGPT-4o (κ = 0.259; %95 CI: 0.100, 0.418), while moderate agreement was observed between the first and second round results of Claude 3.5 Sonnet (κ = 0.524; %95 CI: 0.404, 0.644).

In the evaluation of the AIS localization in specific lobes or regions, moderate agreement was found between the first and second round results of ChatGPT-4o (κ = 0.443; %95 CI: 0.343, 0.543), while fair agreement was found between the results of Claude 3.5 Sonnet (κ = 0.384; %95 CI: 0.284, 0.484).

## 4. Discussion

In this study, we evaluated the ability of the ChatGPT-4o and Claude 3.5 Sonnet AI models to detect the presence and localization of AIS from DWI and ADC images. To the best of our knowledge, this is the first study in the literature that uses both ChatGPT-4o and Claude 3.5 Sonnet for AIS detection.

In this study, while Claude 3.5 Sonnet achieved a high diagnostic accuracy rate (84.5%), ChatGPT-4o had a low diagnostic accuracy rate (51.8%). No statistically significant difference was found between ChatGPT-4o’s predictions and random guessing (*p* > 0.05). This suggests that ChatGPT-4o was making random predictions in image evaluation. In the study by Kuzan et al., ChatGPT showed a sensitivity of 79.57% and specificity of 84.87% in stroke detection, whereas, in our study, ChatGPT-4o exhibited a higher sensitivity (100%) but a significantly lower specificity (3.6%) [25]. Kuzan et al. excluded lacunar infarcts smaller than 1 cm in their study [25]. However, in our study, AIS cases of all sizes were included, which could affect the sensitivity and specificity values of the models. There are no existing literature data to compare the performance of Claude 3.5 Sonnet, and it is anticipated that our study will serve as a reference for future research.

In detecting the hemispheric localization of AIS, Claude 3.5 Sonnet (67.3%) demonstrated a statistically significantly better performance than ChatGPT-4o (32.3%) (*p* = 0.0006). ChatGPT-4o’s low hemispheric localization success (32.3%) is similar to the findings in Kuzan et al.’s study (26.2%) [25]. Both models showed the lowest success in detecting the affected specific lobe or region. However, Claude 3.5 Sonnet demonstrated a statistically significantly higher performance compared to ChatGPT-4o (30.9% vs. 7.3%, *p* = 0.0016). In Kuzan et al.’s study, the lowest success rate (20.4%) was also observed in detecting the specific lobe or region [25]. These results indicate that Claude 3.5 Sonnet outperformed ChatGPT-4o in localizing AIS. Additionally, our study reveals that both models show performance limitations in situations requiring complex assessments. The poor performance of models in detecting AIS localizations significantly limits their clinical use at present. In a clinical setting, the accurate localization of findings is crucial for determining correct diagnosis, treatment, and management strategies. Localization errors—for example, a false positive in an area without pathology or a false negative in an area with pathology—can adversely affect patient outcomes by leading to incorrect or delayed clinical interventions.

In terms of detecting the presence of AIS, ChatGPT-4o showed slight agreement with the radiologists’ gold-standard assessments (κ = 0.036; %95 CI: −0.013, 0.085), while Claude 3.5 Sonnet demonstrated substantial agreement (κ = 0.691; %95 CI: 0.558, 0.824). Both ChatGPT-4o and Claude 3.5 Sonnet showed moderate agreement in their internal evaluations regarding the presence of AIS (κ = 0.428; %95 CI: 0.026, 0.830 and κ = 0.582; %95 CI: 0.433, 0.731, respectively). When assessing the hemispheric localization of AIS, ChatGPT-4o demonstrated fair internal agreement (κ = 0.259; %95 CI: 0.100, 0.418), while Claude 3.5 Sonnet showed moderate internal agreement (κ = 0.524; %95 CI: 0.404, 0.644). In evaluating the specific lobe or region localization of AIS, ChatGPT-4o exhibited moderate internal agreement (κ = 0.443; %95 CI: 0.343, 0.543), whereas Claude 3.5 Sonnet showed fair internal agreement (κ = 0.384; %95 CI: 0.284, 0.484). These findings indicate that the AI models demonstrate some level of consistency in AIS detection, with Claude 3.5 Sonnet showing relatively higher internal consistency. However, these results also suggest that both models need further development to optimize their diagnostic consistency.

LLMs like ChatGPT and Claude have stochastic output generation mechanisms [25]. This characteristic leads to different results being produced in repeated processes for the same input. This stochastic nature has the potential to enhance user experience by increasing diversity and dynamism in human–machine interactions. However, this becomes a significant limiting factor when applied in areas that require precision, such as medical image analysis. Since consistency and repeatability are critical in medical diagnosis and interpretation processes, this structural feature of the models limits their reliability and validity in clinical applications. Both models produced false positive and false negative results, and the stochastic nature of the models may be one of the reasons for these outcomes. Additionally, the lack of specific training for medical image analysis and the limited image processing capabilities of these models could also contribute to the incorrect results.

The use LLMs, especially ChatGPT, in radiology is steadily increasing. In the study by Akinci D’Antonoli and colleagues, both the benefits and challenges of using language models in radiology are highlighted [27]. Recent literature shows that these models are capable of tasks such as data mining from free-text radiology reports [28], generating structured reports [29], answering disease-related questions [30], and responding to radiology-board-style exam questions [31]. Furthermore, recent studies suggest that evaluating radiological images with these models may be feasible [23,24,32]. LLMs could assist radiologists in image assessment and clinicians in diagnosing at centers without radiologists, but it should be noted that these models are still in the early stages and can make errors.

Our study demonstrates that LLMs, such as ChatGPT and Claude, still face significant limitations in their application to medical image analysis. It was observed that neither model, particularly ChatGPT, achieved sufficient diagnostic sensitivity and accuracy in detecting AIS. ChatGPT’s notably high false-positive results in the healthy control group and its predictions resembling random guesses pose a risk of unnecessary additional testing and treatment, raising significant clinical concerns.

Nevertheless, our study highlights the potential of these models in detecting AIS from DW-MRI images, contributing significantly to the theoretical and practical knowledge base in the fields of AI and neuroradiology. Despite their current inability to achieve the desired success in medical image analysis, we anticipate that technological advancements will enable these models to reach high success rates in the coming years, playing a crucial role in optimizing the increasing workload of radiologists. In healthcare centers with insufficient radiologist staffing, healthcare personnel with limited clinical experience could utilize these LLMs for the preliminary evaluation of DW-MRI images, thereby expediting diagnostic processes. This innovative technology could also serve as a secondary control mechanism for junior radiologists during their professional development, minimizing the risk of overlooking critical findings and enhancing diagnostic reliability. Additionally, in studies implementing optimized protocols aimed at reducing door-to-puncture time (DTPT) in conditions requiring early intervention, such as AIS [8], AI models like LLMs could be incorporated into the image evaluation process by relevant specialists. However, it must be acknowledged that these models are still in their early stages and are prone to errors.

AI has found a wide range of applications in the healthcare sector, from drug development processes to remote patient monitoring [14]. While the use of AI in healthcare services offers innovative solutions in medical diagnosis and treatment, this also brings serious ethical concerns. The use of AI in healthcare is based on nine fundamental ethical principles: accountability, autonomy, equity, integrity, non-maleficence, privacy, security, transparency, and trust [33]. The protection of fundamental ethical principles such as autonomy, beneficence, non-maleficence, justice, and privacy is particularly critical in AI applications [34]. AI systems’ influence on medical diagnosis and treatment decisions can affect the informed decision-making processes of patients and healthcare professionals and may limit autonomy. Furthermore, biases in AI algorithms can lead to unfair healthcare services, while non-transparent decision-making processes can cause trust issues. In this context, adopting principles of explainability and transparency, protecting patient privacy, and clarifying responsibility mechanisms should form the cornerstones of ethical AI development and implementation processes in the integration of AI into healthcare services.

The integration of AI technologies into healthcare services is directly related to public attitudes. Despite its developmental potential in this field, the widespread adoption of AI applications requires social acceptance. Trust in the collection of health data and the use of AI in diagnosis and treatment processes is critical for the integration of these technologies into the healthcare system [10]. Recent Pew Research Center surveys show that a large segment of society has concerns about the standalone use of AI in healthcare services [35]. Additionally, surveys reveal that, while public interest in AI use in the healthcare sector is increasing, certain concerns persist [35]. Furthermore, surveys indicate that, while public interest in AI use in the healthcare sector is increasing, certain concerns persist [35]. These concerns are thought to stem from an unfamiliarity with AI technologies, a lack of knowledge, and concerns about data security [10].

Our study has some limitations. The first is that not all DWI and ADC slices from the patients were uploaded to the models. Not analyzing all slices of patients may lead to missing important details, especially in cases where the lesion is small or located in peripheral regions. It may also prevent the complete mapping of the lesion, which can reduce the diagnostic sensitivity of the models. Particularly in cases where the lesion boundaries are not clear or show heterogeneous density, missing slices may lead to false negative results. Additionally, not analyzing all slices may make it difficult to determine the exact localization of the lesion. If the entire set of images had been uploaded, the diagnostic accuracy of both models might have increased.

The second limitation of the study is the necessity of converting the original radiological images in DICOM format to JPEG or PNG formats to enable the analysis by both models. This conversion process may potentially lead to the loss of diagnostically critical image details and metadata, which could negatively affect the analytical capacity of the AI models and compromise the reliability of the results [24].

The third limitation of our study is that only two widely used LLMs (ChatGPT-4o and Claude 3.5 Sonnet) were evaluated for their performance, while others were not assessed. Future studies should evaluate the performance of specialized medical large vision language models (LVLMs) such as BiomedCLIP and LLaVA-Med, which could potentially demonstrate superior diagnostic capabilities in detecting AIS on DW-MRI images, as well as other emerging LLMs.

Compared to deep-learning-based studies, our small sample size can be considered another limitation of our study. Future studies could evaluate the performance of the models using larger sample sizes.

Another limitation of our study is the potential selection bias that may have occurred during the image selection process for DWI and ADC. The specific inclusion and exclusion criteria applied by the radiologists could have impacted the representativeness of the sample. Finally, the fact that the models were not specifically trained for medical image analysis can be considered another limitation of our study.

## 5. Conclusions

In light of the findings of our study, advanced language models such as ChatGPT and Claude AI can be considered as potential supportive tools in detecting the presence of AIS from DWI and ADC images. The low accuracy rates in localization detection highlight the need for the further development of these models in this regard. The performance demonstrated by these models complements traditional radiological evaluation processes and represents an innovative approach that could be integrated into clinical decision-making mechanisms. However, before these technologies are routinely used in clinical practice, they need to be validated and optimized through larger-scale, multicenter studies. Future studies should evaluate the overall diagnostic performance of these models using larger and more diverse datasets, and particularly focus on improving anatomical localization accuracy.

## Figures and Tables

**Figure 1 jcm-14-00571-f001:**
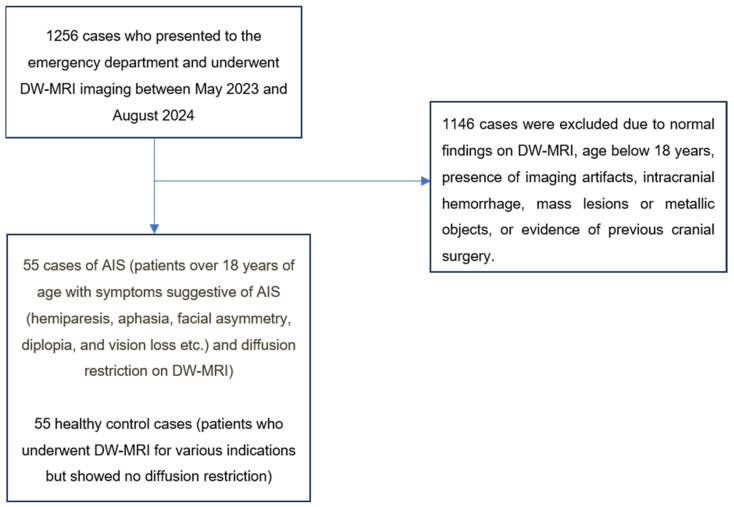
Patient selection process.

**Figure 2 jcm-14-00571-f002:**
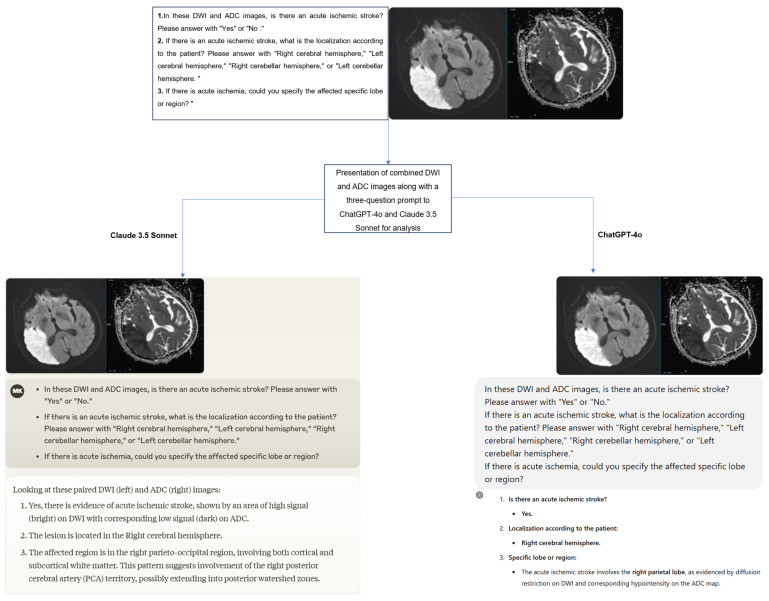
The process of presenting and evaluating combined DWI and ADC images in a single image with a three-question prompt to the models.

**Table 1 jcm-14-00571-t001:** Summary of diagnostic performance metrics for deep-learning-based AI models in stroke diagnosis.

Author and Year	AI Model Used	DL-Based Approaches	Sample Size	Main Findings
Shinohara et al., 2020 [18]	Deep learning	DCNN	702 patients	AUC: 0.87 Accuracy: 0.82 Sensitivity: 0.82 Specificity: 0.81
Cui et al., 2021 [19]	Deep learning	DeepSym-3D-CNN	190 patients	AUC: 0.86 Accuracy: 0.85 Sensitivity: 0.85 Specificity: 0.84
Abedi et al., 2021 [20]	Deep learning	ANN	260 patients	Precision: 0.92 Sensitivity: 0.80 Specificity: 0.86
Zhang et al., 2018 [21]	Deep learning	3D-CNN	242 patients	Dice similarity coefficient: 0.79 Precision: 0.92

AI: artificial intelligence; ANN: artificial neural network; DCNN: deep convolutional neural network; AUC: area under the curve.

**Table 2 jcm-14-00571-t002:** Demographic data of AIS group and healthy control group cases.

	AIS Group (*n* = 55)	Healthy Control Group (*n* = 55)	*p*-Value
Gender ¶, n (%)			
Female	25 (45.5%)	25 (45.5%)	1.0
Male	30 (54.5%)	30 (54.5%)	
Age *, years, Mean ± SD	73.11 ± 11.43	72.75 ± 10.75	0.90

¶ Chi-square Test; * Independent-Samples *t*-test; AIS: acute ischemic stroke; SD: standard deviation.

**Table 3 jcm-14-00571-t003:** Results of ChatGPT-4o and Claude 3.5 Sonnet in assessing the presence of AIS.

	True Positive (n)	False Positive (n)	True Negative (n)	False Negative (n)	Sensitivity (%)	Specificity (%)	PPV (%)	NPV (%)	Diagnostic Accuracy (%)	F1 Score (%)
ChatGPT-4o	55	53	2	0	100	3.6	50.9	100	51.8	67.5
Claude 3.5 Sonnet	52	14	41	3	94.5	74.5	78.8	93.2	84.5	85.9

PPV: positive predictive value; NPV: negative predictive value.

**Table 4 jcm-14-00571-t004:** Comparative analysis of full-accuracy responses given by ChatGPT-4o and Claude 3.5 Sonnet to all questions in the AIS group.

	Completely Correct, n (%)	Incorrect or at Least One Answer Correct, n (%)	Total, n (%)
ChatGPT-4o	4 (7.3)	51 (92.7)	55 (100)
Claude 3.5 Sonnet	17 (30.9)	38 (69.1)	55 (100)

**Table 5 jcm-14-00571-t005:** Evaluation of agreement between ChatGPT-4o and Claude 3.5 Sonnet with radiologists in detecting the presence of AIS.

		Radiologists		
		Negative (n)	Positive (n)	Total (n)
ChatGPT-4o	Negative (n)	2	0	2
	Positive (n)	53	55	108
	Total (n)	55	55	110
Cohen’s Kappa	(κ = 0.036; %95 CI: −0.013, 0.085)			
Claude 3.5 Sonnet	Negative (n)	41	3	44
	Positive (n)	14	52	66
	Total (n)	55	55	110
Cohen’s Kappa	(κ = 0.691; %95 CI: 0.558, 0.824)			

**Table 6 jcm-14-00571-t006:** ChatGPT-4o’s performance in hemispheric localization assessment.

AIS Localization	Correct Answer (n)	Incorrect Answer (n)	Total (n)
No AIS	2	53	55
Right cerebral	16	14	30
Left cerebral	2	21	23
Both cerebral	0	1	1
Left cerebellar	0	1	1
Total (n)	20	90	110

**Table 7 jcm-14-00571-t007:** Claude 3.5 Sonnet’s performance in hemispheric localization assessment.

AIS Localization	Correct Answer (n)	Incorrect Answer (n)	Total (n)
No AIS	41	14	55
Right cerebral	17	13	30
Left cerebral	20	3	23
Both cerebral	0	1	1
Left cerebellar	0	1	1
Total (n)	78	32	110

**Table 8 jcm-14-00571-t008:** Specific lobe-region localizations of AIS accurately identified by ChatGPT-4o and Claude 3.5 Sonnet.

ChatGPT-4o	Correct Answer (n)	Claude 3.5 Sonnet	Correct Answer (n)
Right parietal lobe	2	Right frontal	2
Right periventricular WM	1	Right parietal	5
Left frontal lobe	1	Right temporal	1
		Right temporo-occipital	1
		Left parietal	3
		Left occipital	1
		Left parieto-occipital	2
		Left temporo-occipital	1
		Left thalamus	1
Total (n)	4		17

WM: White matter.

**Table 9 jcm-14-00571-t009:** First and second round evaluation results of ChatGPT-4o and Claude 3.5 Sonnet in detecting the presence of AIS.

	True Positive (n)	False Positive (n)	True Negative (n)	False Negative (n)	Sensitivity (%)	Specificity (%)	PPV (%)	NPV (%)	Diagnostic Accuracy (%)
1. ChatGPT-4o	55	53	2	0	100	3.6	50.9	100	51.8
2. ChatGPT-4o	55	48	7	0	100	12.7	53.4	100	56.4
1. Claude 3.5 Sonnet	52	14	41	3	94.5	74.5	78.8	93.2	84.5
2. Claude 3.5 Sonnet	46	9	46	9	83.6	83.6	83.6	83.6	83.6

## Data Availability

The data presented in this study are available upon request from the corresponding author. The data are not publicly available due to privacy.

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
