# Peer review of "Evaluation of Advanced Artificial Intelligence Algorithms’ Diagnostic Efficacy in Acute Ischemic Stroke: A Comparative Analysis of ChatGPT-4o and Claude 3.5 Sonnet Models"

_jcm, 2025, doi:10.3390/jcm14020571_

Round 1
Reviewer 1 Report
Comments and Suggestions for Authors
The authors evaluate two large vision-language models (LVLMs) in terms of stroke diagnosis. I am surprised to see that Claude achieved such an amazing performance. The accuracy on detecting the existence of stroke and the hemispheric location is great. The experiment results are also solid with kappa coefficient.
I am glad to list my suggestions on further improvement.
1. The results of chatgpt-4o show some concerns. It seems that it simply predicts almost 99% cases as positive, and randomly predict their locations. Could you do a test and see how its predictions are close to random guess?
2. Comparing the performance of two LVLMs is not very meaningful. By now, there have been many medical LVLMs which i believe have better performance than chatgpt-4o and claude. For example, BiomedCLIP and LLaVA-Med. The trained models are accessible online and you may also test their performance.
3. chatgpt-4o and claude are not specific for medical questions. Have you thought about fine-tuning LVLMs on stroke DW-MRIs? In this case, they may achieve better performance.
4. Why it is meaningful to test the zero-shot performance of general LVLMs on medical questions, given they can be fine-tuned on medical datasets to have better accuracy?
Author Response
|
Dear Reviewer, We sincerely thank you for thoroughly reviewing our study and providing constructive feedback. We have carefully evaluated all your comments and suggestions and revised the manuscript accordingly. All changes made in the revised manuscript have been marked using the track changes feature, allowing you to easily follow the corrections. Below, you can find our detailed responses to each of your suggestions and the corresponding changes we made in the manuscript listed as bullet points. For some of your suggestions, there were points that we could not fully implement due to time constraints and methodological limitations of our study. We have explained these situations with their justifications and noted them as valuable suggestions for future studies. We thank you again for your contributions to the review process and your constructive approach. If you think any additional clarification or correction is needed, please do not hesitate to let us know. Best regards.
Comments 1. The results of chatgpt-4o show some concerns. It seems that it simply predicts almost 99% cases as positive, and randomly predict their locations. Could you do a test and see how its predictions are close to random guess?
Response 1: Following your suggestion, we conducted a binomial test to determine how much the models differed from random guessing. We included the results in the Results section of our manuscript and discussed them in the Discussion section. Your contribution was incredibly valuable to us, and we sincerely thank you. The statement we added to the results section of our manuscript is as follows: The binomial test revealed that ChatGPT-4o's accuracy in detecting AIS (51.8%) was not statistically different from random guessing (50%) (p = 0.775). In contrast, the same test demonstrated that Claude 3.5 Sonnet's accuracy in detecting AIS was 84.5%, which is significantly higher than the random guessing rate of 50% (p = 0.000). The statement we added to the discussion section of our manuscript is as follows: In this study, while Claude 3.5 Sonnet achieved a high diagnostic accuracy rate (84.5%), ChatGPT-4o had a low diagnostic accuracy rate (51.8%). No statistically significant difference was found between ChatGPT-4o's predictions and random guessing (p>0.05). This suggests that ChatGPT-4o was making random predictions in image evaluation.
Comment 2 Comparing the performance of two LVLMs is not very meaningful. By now, there have been many medical LVLMs which i believe have better performance than chatgpt-4o and claude. For example, BiomedCLIP and LLaVA-Med. The trained models are accessible online and you may also test their performance.
Response 2: Dear Reviewer, Thank you for your valuable suggestions regarding artificial intelligence models. We share your view that domain-specific large vision language models (LVLMs) like BiomedCLIP and LLaVA-Med, which are specifically designed for medical applications, may demonstrate superior performance. However, the primary aim of our study was to evaluate the potential and limitations of two "general-purpose" large language models (LLMS) (ChatGPT and Claude) that are widely used in acute ischemic stroke diagnosis. Therefore, our research question and data collection process were designed to compare these two models specifically. When planning our study, we submitted our ethics committee application specifically for ChatGPT and Claude models. We established the necessary approval and data processing protocol for the use of these models. Consequently, including different models would have fallen outside the scope of the ethics committee approval. We do not disregard the potential advantages of medical domain-specific LVLMs (such as BiomedCLIP and LLaVA-Med). On the contrary, we believe that future studies involving these models could enrich the findings obtained from our research. Investigating the clinical benefits of these models in critical conditions like acute ischemic stroke could be our next step. Evaluating the performance of the models you suggested will be a valuable research topic for our future studies. We have added this point to the limitations section of our study and emphasized the importance of evaluating the performance of other LLMs and specialized medical LVLMs like BiomedCLIP and LLaVA-Med in the future research recommendations section. Here is our added sentence to the limitations section of our study. The third limitation of our study is that only two widely used LLMs (ChatGPT-4o and Claude 3.5 Sonnet) were evaluated for their performance, while others were not assessed. Future studies should evaluate the performance of specialized medical large vision language models (LVLMs) such as BiomedCLIP and LLaVA-Med, which could potentially demonstrate superior diagnostic capabilities in detecting AIS on DW-MRI images, as well as other emerging LLMs.
Comment 3 Chatgpt-4o and claude are not specific for medical questions. Have you thought about fine-tuning LVLMs on stroke DW-MRIs? In this case, they may achieve better performance. Response 3: Dear reviewer, to the best of our knowledge and based on our research, there is currently no publicly available, comprehensive fine-tuning option for ChatGPT or Claude. Therefore, we were unfortunately unable to implement fine-tuning on DW-MRI images in our study. If you are aware of any fine-tuning options that we may not know about, we would be very happy to receive your feedback. Comment 4. Why it is meaningful to test the zero-shot performance of general LVLMs on medical questions, given they can be fine-tuned on medical datasets to have better accuracy?
Response 4: In our study, we considered it important to understand how widely accessible LLMs perform without any additional training or customization, as this would reveal their real-world usage potential and possible limitations. Therefore, in our study, we chose to focus on the zero-shot performance of these general-purpose large language models.
We are grateful for the time you have dedicated to our study. Your recommendations have guided us in preparing a more enhanced and structured version of our study. We hope that our revisions meet your expectations. Sincerely
|

Reviewer 2 Report
Comments and Suggestions for Authors
This paper suggested Evaluation of Advanced Artificial Intelligence Algorithms' Diagnostic Efficacy in Acute Ischemic Stroke: A Comparative Analysis of ChatGPT-4o and Claude 3.5 Sonnet Models. These findings underline the potential of AI tools in radiological diagnostics while acknowledging their current limitations. I suggest a major revision based on the following comments:
1. The introduction is too short. please provide the research gap including more clarifications in introduction part.
2. Please add the structure of your paper at the end of the introduction including more details of each sections and subsection.
3. Please add the literature review part after introduction.
4. Patient selection process in Figure 1 needs more details and explanations.
5. Please improve the quality of Figure 2. It is something like a picture.
6. Subsection 2.7. Statistical analysis needs more explanations and references such as including NNOVA.
7. AIS: Acute Ischemic Stroke needs more clarification and relevant references. Please add "A data envelopment analysis model for optimizing transfer time of ischemic stroke patients under endovascular thrombectomy. Healthcare Analytics. 2024 Dec 1;6:100364" , "Streamlined triage and transfer protocols improve door-to-puncture time for endovascular thrombectomy in acute ischemic stroke"
8. Please add limitation and future studies at the end of conclusion.
9. Only 20 references are too low. Please add more relevant MDPI following references:
- Selected mediators of inflammation in patients with acute ischemic stroke. International Journal of Molecular Sciences. 2022 Sep 13;23(18):10614.
Comments on the Quality of English LanguageThe English could be improved to more clearly express the research.
Author Response
Dear Reviewer,
We sincerely thank you for thoroughly reviewing our study and providing constructive feedback. We have carefully evaluated all your comments and suggestions and revised the manuscript accordingly.
All changes made in the revised manuscript have been marked using the track changes feature, allowing you to easily follow the corrections. Below, you can find our detailed responses to each of your suggestions and the corresponding changes we made in the manuscript listed as bullet points.
For some of your suggestions, there were points that we could not fully implement due to time constraints and methodological limitations of our study. We have explained these situations with their justifications and noted them as valuable suggestions for future studies.
We thank you again for your contributions to the review process and your constructive approach. If you think any additional clarification or correction is needed, please do not hesitate to let us know.
Best regards.
Comment 1.
The introduction is too short. please provide the research gap including more clarifications in introduction part.
Response 1:
Based on your suggestion, we have expanded the introduction section of our manuscript. The newly added paragraphs are presented below.
It is reported that stroke is more common in elderly individuals, with the average age being 69 years (2).
Patients in the post-stroke period typically experience physical movement limitations, communication disorders, and cognitive function losses (6). These multiple impairments cause patients to require comprehensive rehabilitation and intensive medical care. When AIS is intervened in the early period, it is a treatable condition, and therefore its early detection is critically important (7). AIS can be treated with a medical approach called "thrombolysis" using alteplase (also known as tissue Plasminogen Activator (tPA)) or Tenecteplase (TNK) (2). Another treatment option for AIS is mechanical thrombectomy, and studies are being conducted aiming to reduce the door-to-puncture time (DTPT) to enable rapid intervention (8).
However, recent studies suggest that non-contrast computed tomography (NCCT) and computed tomography angiography (CTA), which can be performed more quickly than DW-MRI, should be primarily implemented to detect AIS and rule out intracranial hemorrhage (ICH) so that treatment can be initiated without delay (2,8).
The use of AI in healthcare covers a broad area (10). CardioAI, which analyzes cardiac magnetic resonance images, and computer-aided diagnosis (CAD) systems used in endoscopy-colonoscopy procedures (11) can be given as examples of AI applications in healthcare.
AI technology is divided into various specialized subdisciplines, including machine learning (ML), deep learning (DL), and computer vision (11). ML identifies patterns using specific features and optimizes decision-making processes by applying this knowledge to similar situations. This technology has evolved into DL, which utilizes artificial neural networks (ANN) mimicking human neural systems. Image processing technology encompasses the digital perception and analysis of visual data (11).
ChatGPT-4o (ChatGPT-4 Omni) and Claude 3.5 Sonnet models have been recently devel-oped and offer the ability to evaluate visual materials.
Recent studies demonstrate the effectiveness of DL applications in detecting various pathologies from radiological imaging data (15-17). DL applications are also coming to the forefront in AIS detection (18-21). It is reported that sensitivity and specificity values above 80% have been achieved with these AI applications. However, there are only a few studies on the detection of pathologies from radiological images using ChatGPT (22-25), and among these, only the study by Kuzan et al. focused on AIS detection from DW-MRI images (25).
Comment 2.
Please add the structure of your paper at the end of the introduction including more details of each sections and subsection.
Response 2:
Thank you for your suggestion. However, the methodological structure of our article and its section transitions are already consistent with similar studies in the literature and written in the standard IMRAD (Introduction, Methods, Results and Discussion) format. Additionally, each section has a logical flow with its own subheadings. We believe that adding a structural summary at the end of the Introduction section might disrupt the fluidity and integrity of the article. Therefore, if you agree, we think it would be more appropriate to maintain the current format rather than providing an additional structural summary at the end of the introduction section.
Comment 3.
Please add the literature review part after introduction.
Response 3:
Based on your suggestion, we have added a literature review section after the introduction. The new section and the table summarizing the literature are as follows.
Recent studies demonstrate the effectiveness of DL applications in detecting various pa-thologies from radiological imaging data [15-17]. DL applications are also coming to the forefront in AIS detection (Table 1) [18-21]. It is reported that sensitivity and specificity values above 80% have been achieved with these AI applications. However, there are only a few studies on the detection of pathologies from radiological images using ChatGPT [22-25], and among these, only the study by Kuzan et al. focused on AIS detection from DW-MRI images [25].
Table 1. Summary of diagnostic performance metrics for deep learning-based AI models in stroke diagnosis.
|
Author and Year |
AI Model Used |
DL-based approaches |
Sample Size |
Main Findings |
|
Shinohara et al., 2020 (18) |
Deep learning |
DCNN |
702 patients |
AUC: 0.87 Accuracy: 0.82 Sensitivity: 0.82 Specificity: 0.81 |
|
Cui et al., 2021 (19) |
Deep learning |
DeepSym-3D-CNN |
190 patients |
AUC: 0.86 Accuracy: 0.85 Sensitivity: 0.85 Specificity: 0.84 |
|
Abedi et al., 2021 (20) |
Deep learning |
ANN |
260 patients |
Precision: 0.92 Sensitivity: 0.80 Specificity: 0.86 |
|
Zhang et al., 2018 (21) |
Deep learning |
3D-CNN |
242 patients |
Dice similarity coefficient: 0.79 Precision: 0.92 |
AI: artificial intelligence; ANN: artificial neural network; DCNN: deep convolutional neural network; AUC: area under the curve
Comment 4.
Patient selection process in Figure 1 needs more details and explanations.
Response 4:
Following your suggestion, we have revised the patient selection process in more detail. Figure 1's explanations were revised. The revised version of the patient selection and Figure 1 is as follows:
A total of 1256 patients with DW-MRI images recorded in our hospital's information management system between May 2024 and September 2024 were retrospectively analyzed. These patients underwent systematic evaluation. Accordingly, patients over 18 years of age who presented to the emergency department with neurological symptoms suggestive of AIS (hemiparesis, aphasia, facial asymmetry, diplopia, and vision loss) and underwent DW-MRI on the same day with preliminary diagnosis of AIS were identified. Among the screened patients, 55 cases with diffusion restriction on DW-MRI were classified as the AIS group. Cases with clinical findings but no diffusion restriction on DW-MRI were not included in the AIS group (Figure 1).
The control group consisted of 55 healthy cases who underwent DW-MRI for various indications but showed no diffusion restriction (Figure 1).
The exclusion criteria for both groups were standardized as follows: the presence of artifacts limiting image evaluation, age under 18 years, a history of intracranial mass or recent cranial surgery, intracranial hemorrhage, or the presence of metallic objects (Figure 1).
Cases were determined by consensus between two radiologists with 8 and 12 years of experience.

Figure 1. Patient selection process
Comment 5.
Please improve the quality of Figure 2. It is something like a picture.
Response 5:
Based on your suggestion, we improved the quality of Figure 2 and made several revisions. The new figure is as follows.

Figure 2. The process of combining DWI and ADC images into a single image and their evaluation by the models.
Comment 6.
Subsection 2.7. Statistical analysis needs more explanations and references such as including NNOVA.
Response 6:
Dear Reviewer,
Based on your suggestions, we have added more details to the statistical analysis section of the manuscript. We assumed that by "NNOVA test," you were referring to the ANOVA test. Since we compared both models with the gold standard (radiologist responses) and with each other, and did not evaluate more than two groups, we did not use the ANOVA test in our statistical analyses. If you agree, we believe it would be appropriate not to include this test in the statistical analysis section.
Here are the added statistical analysis methods:
The effect of the age variable on the models' AIS detection performance was evaluated using binary logistic regression analysis, while the effect of the gender variable on model performance was assessed using the Pearson Chi-square test. A binomial test was used to compare the models' predictions with random guessing.
Comment 7.
AIS: Acute Ischemic Stroke needs more clarification and relevant references. Please add "A data envelopment analysis model for optimizing transfer time of ischemic stroke patients under endovascular thrombectomy. Healthcare Analytics. 2024 Dec 1;6:100364" , "Streamlined triage and transfer protocols improve door-to-puncture time for endovascular thrombectomy in acute ischemic stroke"
Response 7:
Based on your suggestion, we have added the references you indicated to the introduction and discussion sections of our manuscript. The added paragraphs are as follows.
İntroduction
It is reported that stroke is more common in elderly individuals, with the average age being 69 years (2).
AIS can be treated with a medical approach called "thrombolysis" using alteplase (also known as tissue Plasminogen Activator (tPA)) or Tenecteplase (TNK) (2).
Another treatment option for AIS is mechanical thrombectomy, and studies are being conducted aiming to reduce the door-to-puncture time (DTPT) to enable rapid intervention [8].
However, recent studies suggest that non-contrast computed tomography (NCCT) and computed tomography angiography (CTA), which can be performed more quickly than DW-MRI, should be primarily implemented to detect AIS and rule out intracranial hemorrhage (ICH) so that treatment can be initiated without delay (2,8).
Discussion
Additionally, in studies implementing optimized protocols aimed at reducing DTPT in conditions requiring early intervention like AIS (8), AI models such as LLMs can be involved in the image evaluation step by relevant specialists.
Comment 8.
Please add limitation and future studies at the end of conclusion.
Response 8:
Based on your suggestion, we would like to note that we have added the following paragraph to the conclusion section of our manuscript:
Future studies should evaluate the overall diagnostic performance of these models using larger and more diverse datasets, and particularly focus on improving anatomical localization accuracy.
Comment 9.
Only 20 references are too low. Please add more relevant MDPI following references: - Selected mediators of inflammation in patients with acute ischemic stroke. International Journal of Molecular Sciences. 2022 Sep 13;23(18):10614.
Response 9:
Based on your suggestion, we have added the reference you indicated (reference 6) to the introduction section. Additionally, we expanded the introduction and discussion sections of our manuscript and increased our number of references to 35. The reference you suggested is specified below.
Patients in the post-stroke period typically experience physical movement limitations, communication disorders, and cognitive function losses (6). These multiple impairments cause patients to require comprehensive rehabilitation and intensive medical care.
We are grateful for the time you have dedicated to our study. Your recommendations have guided us in preparing a more enhanced and structured version of our study. We hope that our revisions meet your expectations.
Sincerely

Reviewer 3 Report
Comments and Suggestions for Authors
The study only compared the performance of ChatGPT-4o and Claude 3.5 Sonnet, two language models. However, there are numerous other models available that could have been considered for this study. I believe that a more comprehensive comparison involving a larger number (~10) of different models would have provided a more robust analysis of the current state of AI in AIS detection. By limiting the comparison to only two models, the study may not fully capture the variability and potential of different AI architectures in this specific task.
You considered a 'true positive' solely based on the correct answer to the first question about the presence of an acute ischemic stroke (is there an acute ischemic stroke?), without taking into account the accuracy of the localization questions (where is it?). I believe this approach is flawed and may lead to an overestimation of the models' diagnostic performance. In my opinion, it cannot be considered a 'true positive' if the model wrongly classified the image as having an acute ischemic stroke in the wrong location. If the model detects an acute ischemic stroke where it does not exist, it indicates a misdiagnosis, not a correct identification. Therefore, I argue that a 'true positive' should only be counted when the model correctly identifies both the presence and the proper location of the acute ischemic stroke. While you can keep the results as you have them, I want to see another set of results that show the confusion matrices and resulting accuracy when 'true positive' is only considered when the acute ischemic stroke is properly located. This additional analysis would provide a more comprehensive and accurate assessment of the models' diagnostic capabilities.
Provide more details on the specific versions of the AI models used (e.g., exact release dates or version numbers).
Clarify the process of converting DICOM images to JPEG format, including any potential image quality adjustments made during this process.
Specify the exact software and settings used for image anonymization.
More clearly delineate the strengths and limitations into separate subsections for easier readability.
Discuss the potential impact of not using all DWI and ADC slices on the models' performance in greater detail.
Address the lack of specific training for medical image analysis as a limitation of the study.
Provide more statistical analysis, such as confidence intervals, to support the significance of the differences between the models' performance.
Address the potential impact of the models' poor localization accuracy on their clinical utility. Again, if localized somewhere where it isn't, that's not a 'true positive'.
Discuss public perspectives on using AI in healthcare with recent references (2024).
Discuss the ethical landscape of using AI in healthcare, again, with recent references (2024).
Author Response
Dear Reviewer,
We sincerely thank you for thoroughly reviewing our study and providing constructive feedback. We have carefully evaluated all your comments and suggestions and revised the manuscript accordingly.
All changes made in the revised manuscript have been marked using the track changes feature, allowing you to easily follow the corrections. Below, you can find our detailed responses to each of your suggestions and the corresponding changes we made in the manuscript listed as bullet points.
For some of your suggestions, there were points that we could not fully implement due to time constraints and methodological limitations of our study. We have explained these situations with their justifications and noted them as valuable suggestions for future studies.
We thank you again for your contributions to the review process and your constructive approach. If you think any additional clarification or correction is needed, please do not hesitate to let us know.
Best regards.
Comment 1.
The study only compared the performance of ChatGPT-4o and Claude 3.5 Sonnet, two language models. However, there are numerous other models available that could have been considered for this study. I believe that a more comprehensive comparison involving a larger number (~10) of different models would have provided a more robust analysis of the current state of AI in AIS detection. By limiting the comparison to only two models, the study may not fully capture the variability and potential of different AI architectures in this specific task.
Response 1:
Dear Reviewer,
First of all, thank you for your comprehensive suggestions regarding artificial intelligence models. The main purpose of our study was to demonstrate the potential and limitations of two widely used "general-purpose" large language models (LLMs) (ChatGPT and Claude) in acute ischemic stroke diagnosis. Therefore, our research question and data collection process were designed to compare these two models. When planning our study, we submitted our ethics committee application specifically for ChatGPT and Claude models. We established the necessary approval and data processing protocol for the use of these models. Consequently, including different models would fall outside the scope of the ethics committee approval. Our study aimed to encourage attempts to analyze radiological images with LLMs such as ChatGPT and Claude and to serve as a methodological reference point. Evaluating the performance of the other models you suggested will be a valuable research topic for our future studies. We have added this point to the limitations section of our study and emphasized the importance of evaluating the performance of other LLMs and specialized medical LVLMs like BiomedCLIP and LLaVA-Med in the future research recommendations section.
Here is our added sentence to the limitations section of our study.
The third limitation of our study is that only two widely used LLMs (ChatGPT-4o and Claude 3.5 Sonnet) were evaluated for their performance, while others were not assessed. Future studies should evaluate the performance of specialized medical large vision language models (LVLMs) such as BiomedCLIP and LLaVA-Med, which could potentially demonstrate superior diagnostic capabilities in detecting AIS on DW-MRI images, as well as other emerging LLMs.
Comment 2.
You considered a 'true positive' solely based on the correct answer to the first question about the presence of an acute ischemic stroke (is there an acute ischemic stroke?), without taking into account the accuracy of the localization questions (where is it?). I believe this approach is flawed and may lead to an overestimation of the models' diagnostic performance. In my opinion, it cannot be considered a 'true positive' if the model wrongly classified the image as having an acute ischemic stroke in the wrong location. If the model detects an acute ischemic stroke where it does not exist, it indicates a misdiagnosis, not a correct identification. Therefore, I argue that a 'true positive' should only be counted when the model correctly identifies both the presence and the proper location of the acute ischemic stroke. While you can keep the results as you have them, I want to see another set of results that show the confusion matrices and resulting accuracy when 'true positive' is only considered when the acute ischemic stroke is properly located. This additional analysis would provide a more comprehensive and accurate assessment of the models' diagnostic capabilities.
Response 2:
Dear reviewer, we would like to note that we have added Table 8 in accordance with your suggestion. Additionally, we would like to mention that we have added the limitation you pointed out regarding localization to the conclusion section of the manuscript.
Both models answered all the questions in the prompt consistently with each other. When the models responded that AIS was present, they answered the other questions accordingly, and when they responded that AIS was not present, they left the other questions unanswered. The number of cases where ChatGPT-4o answered all questions with complete accuracy was 4 (7.3%), while for Claude 3.5 Sonnet it was 17 (30.9%) (Table 4).
Table 4. Comparative analysis of full-accuracy responses given by ChatGPT-4o and Claude 3.5 Sonnet to all questions in the AIS group
|
|
Completely correct, n(%) |
Incorrect or at least one answer correct, n(%) |
Total, n(%) |
|
ChatGPT-4o |
4 (7.3) |
51 (92.7) |
55 (100) |
|
Claude 3.5 Sonnet |
17 (30.9) |
38 (69.1) |
55 (100) |
Comment 3.
Provide more details on the specific versions of the AI models used (e.g., exact release dates or version numbers).
Response 3:
In accordance with your suggestion, we added the specific versions and other details of the AI models used as a subheading in the materials and methods section as follows:
AI Models used in the study
In our study, we used ChatGPT-4o (GPT-4 Omni), developed by OpenAI (OpenAI, San Francisco, CA, USA) and introduced on May 13, 2024, and Claude 3.5 Sonnet model, developed by Anthropic (Anthropic, San Francisco, CA, USA) and introduced on June 20, 2024, to analyze DW-MRI images.
Comments 4 and 5.
Clarify the process of converting DICOM images to JPEG format, including any potential image quality adjustments made during this process.
Responses 4 and 5:
We would like to note that, based on your suggestions, we have revised the explanation of the process of converting images from DICOM to JPEG and the software used for image anonymization as follows.
The original DICOM (Digital Imaging and Communications in Medicine) format DWI and ADC images of the cases were first de-identified using AW Volumeshare 7 software (AW version 4.7, GE Healthcare, Milwaukee, WI, USA). Subsequently, the images were converted to JPEG format through this software without altering basic parameters such as resolution, brightness, and contrast. All images were exported while preserving the original pixel size. During conversion, high-level JPEG quality was preferred to preserve diagnostically important details as much as possible. Later, DWI and ADC images were combined into a single image using Microsoft paint software (Microsoft, Corp., Redmond, WA, USA), and irrelevant regions outside the brain were cropped from the newly created image. Additionally, orientation verification was performed by radiologists on the images before uploading them to the models.
Comment 6.
More clearly delineate the strengths and limitations into separate subsections for easier readability.
Response 6:
Based on your suggestion, the strengths and limitations of the study were organized as separate subsections.
Comment 7.
Discuss the potential impact of not using all DWI and ADC slices on the models' performance in greater detail.
Response 7:
We would like to note that we have added the following paragraph to the discussion section regarding the potential impact of not using all DWI and ADC slices on the models' performance:
Not analyzing all slices of patients may lead to missing important details, especially in cases where the lesion is small or located in peripheral regions. It may also prevent complete mapping of the lesion, which can reduce the diagnostic sensitivity of the models. Particularly in cases where the lesion boundaries are not clear or show heterogeneous density, missing slices may lead to false negative results. Additionally, not analyzing all slices may make it difficult to determine the exact localization of the lesion.
Comment 8.
Address the lack of specific training for medical image analysis as a limitation of the study.
Response 8:
Following your suggestion, we noted the lack of specialized training for medical image analysis of the models in the limitations section as follows:
Finally, another limitation of our study is that the models we used were not specifically trained for medical image analysis.
Comment 9.
Provide more statistical analysis, such as confidence intervals, to support the significance of the differences between the models' performance.
Response 9:
Based on your suggestion, we have added confidence (CI) intervals for each data point to support the significance of differences in model performance.
We would like to note that we have expanded the statistical analysis section. Below are the new sentences we have added.
The effect of the age variable on the models' AIS detection performance was evaluated using binary logistic regression analysis, while the effect of the gender variable on model performance was assessed using the Pearson Chi-square test. A binomial test was used to compare the models' predictions with random guessing
Comment 10.
Address the potential impact of the models' poor localization accuracy on their clinical utility. Again, if localized somewhere where it isn't, that's not a 'true positive'.
Response 10:
Based on your suggestion, we have added the following paragraph to the discussion section of our manuscript:
The poor performance of models in detecting AIS localizations significantly limits their clinical use at present. In a clinical setting, accurate localization of findings is crucial for determining correct diagnosis, treatment, and management strategies. Localization errors—for example, a false positive in an area without pathology or a false negative in an area with pathology—can adversely affect patient outcomes by leading to incorrect or delayed clinical interventions.
Comment 11.
Discuss public perspectives on using AI in healthcare with recent references (2024).
Response 11:
Based on your suggestion, we have added the following paragraph about public perspectives on the use of AI in healthcare to the discussion section of our manuscript:
The integration of AI technologies into healthcare services is directly related to public attitudes. Despite its developmental potential in this field, the widespread adoption of AI applications requires social acceptance. Trust in the collection of health data and the use of AI in diagnosis and treatment processes is critical for the integration of these technolo-gies into the healthcare system [10]. Recent Pew Research Center surveys show that a large segment of society has concerns about the standalone use of AI in healthcare services [35]. Additionally, surveys reveal that while public interest in AI use in the healthcare sector is increasing, certain concerns persist [35]. Furthermore, surveys indicate that while public interest in AI use in the healthcare sector is increasing, certain concerns persist (35). These concerns are thought to stem from unfamiliarity with AI technologies, lack of knowledge, and concerns about data security [10].
Comment 12.
Discuss the ethical landscape of using AI in healthcare, again, with recent references (2024).
Response 12:
Based on your suggestion, we have added the following paragraph about the ethical dimension of AI use in healthcare services to the discussion section of our manuscript:
AI has found a wide range of applications in the healthcare sector, from drug development processes to remote patient monitoring (14). While the use of AI in healthcare services offers innovative solutions in medical diagnosis and treatment, this also brings serious ethical concerns. The use of AI in health care is based on nine fundamental ethical principles: accountability, autonomy, equity, integrity, non-maleficence, privacy, security, transparency, and trust (33). The protection of fundamental ethical principles such as autonomy, beneficence, non-maleficence, justice, and privacy is particularly critical in AI applications (34). AI systems' influence on medical diagnosis and treatment decisions can affect the informed decision-making processes of patients and healthcare professionals and may limit autonomy. Furthermore, biases in AI algorithms can lead to unfair healthcare services, while non-transparent decision-making processes can cause trust issues. In this context, adopting principles of explainability and transparency, protecting patient privacy, and clarifying responsibility mechanisms should form the cornerstones of ethical AI development and implementation processes in the integration of AI into healthcare services.
We are grateful for the time you have dedicated to our study. Your recommendations have guided us in preparing a more enhanced and structured version of our study. We hope that our revisions meet your expectations.
Sincerely

Reviewer 4 Report
Comments and Suggestions for Authors
I suggest the following improvements.
Abstract
Include specific performance metrics (e.g., sensitivity, specificity) and their statistical significance to highlight key findings.
Introduction
Streamline general discussions about stroke and AI to focus on the study's objectives.
Methods
Provide more details about the criteria used to select the imaging slices and ensure that data represent variability in clinical presentations.
Results
Add results for specific subgroups, such as age or gender, to understand their potential influence on model performance.
Discussion
Highlight methodological constraints, such as the small sample size or the necessity to convert images to JPEG, and suggest ways to address these in future studies.
Conclusion
Provide explicit guidance on the potential use of these AI models in clinical practice, specifying limitations.
Figures
Improve the resolution of figures to enhance readability.
Author Response
Dear Reviewer,
We sincerely thank you for thoroughly reviewing our study and providing constructive feedback. We have carefully evaluated all your comments and suggestions and revised the manuscript accordingly.
All changes made in the revised manuscript have been marked using the track changes feature, allowing you to easily follow the corrections. Below, you can find our detailed responses to each of your suggestions and the corresponding changes we made in the manuscript listed as bullet points.
For some of your suggestions, there were points that we could not fully implement due to time constraints and methodological limitations of our study. We have explained these situations with their justifications and noted them as valuable suggestions for future studies.
We thank you again for your contributions to the review process and your constructive approach. If you think any additional clarification or correction is needed, please do not hesitate to let us know.
Best regards.
Comment 1.
Abstract
Include specific performance metrics (e.g., sensitivity, specificity) and their statistical significance to highlight key findings.
Response 1:
Dear reviewer, based on your suggestion, we added additional statistical data such as sensitivity, specificity, and Confidence Interval (CI) to the abstract section of our paper. However, the 250-word limit for the abstract restricted us from adding more data. Thank you very much for your contribution.
Comment 2.
Introduction
Streamline general discussions about stroke and AI to focus on the study's objectives.
Response 2:
Dear reviewer, as per your suggestion, we have reorganized the general discussions about stroke and artificial intelligence to focus on the study objectives. Thank you very much for your contribution.
Comment 3.
Methods
Provide more details about the criteria used to select the imaging slices and ensure that data represent variability in clinical presentations.
Response 3:
We believe your suggestion has made a very important contribution to our manuscript.
Based on your suggestion, we have added the following paragraph to the "Image selection, preparation, and presentation to models" subsection in the materials and methods section:
Image selection, preparation and presentation to models
Image slice selection criteria for AIS cases were standardized with a methodological approach. DWI sequence slices showing the ischemic lesion at its maximum size and optimal visibility, along with their corresponding ADC maps, were designated as reference images for primary analysis. Parameters such as similar distribution numbers of anatomical lesions or similar types of clinical presentations were disregarded in image selection.
The selection of image slices for the healthy control group was systematically performed to show anatomical correlation with the reference slices in the AIS group. The spatial localization of image slices was standardized to match the slice levels in AIS cases as closely as possible. This methodological approach aimed to ensure that AI models analyzed equivalent anatomical structures and localizations in both groups.
The original DICOM (Digital Imaging and Communications in Medicine) format DWI and ADC images of the cases were first de-identified using AW Volumeshare 7 software (AW version 4.7, GE Healthcare, Milwaukee, WI, USA). Subsequently, the images were converted to JPEG format through this software without altering basic parameters such as resolution, brightness, and contrast. All images were exported while preserving the original pixel size. During conversion, high-level JPEG quality was preferred to preserve diagnostically important details as much as possible. Later, DWI and ADC images were combined into a single image using Microsoft paint software (Microsoft, Corp., Redmond, WA, USA), and irrelevant regions outside the brain were cropped from the newly created image. Additionally, orientation verification was performed by radiologists on the images before uploading them to the models. To minimize bias, the cases were presented to the models in a randomized order, accompanied by the three-question prompt mentioned earlier. The responses provided by the models were recorded (Figure 2). Additionally, to assess intra-model reliability, all cases were reanalyzed with the same models two weeks later.
Comment 4.
Results
Add results for specific subgroups, such as age or gender, to understand their potential influence on model performance.
Response 4:
Based on your suggestion, we analyzed the effects of gender and age on model performance, and our findings are presented below. We would like to note that we have added these findings to the results section of our manuscript. Thank you very much.
The AIS detection performance of both models does not show any statistically significant differences between genders. Likewise, no statistically significant differences were found between age and the models' AIS detection performance.
Comment 5.
Discussion Highlight methodological constraints, such as the small sample size or the necessity to convert images to JPEG, and suggest ways to address these in future studies.
Response 5:
Based on your suggestion, in the limitations section of the manuscript, it is stated that the DICOM images had to be converted to JPEG format before being presented to the models, which represents a limitation. Additionally, we would like to note that we have added your suggestion regarding the small sample size to the limitations section as follows.
Second limitation of the study is the necessity of converting the original radiological images in DICOM format to JPEG or PNG formats to enable analysis by both models. This conversion process may potentially lead to the loss of diagnostically critical image details and metadata, which could negatively affect the analytical capacity of the AI models and compromise the reliability of the results (24).
Compared to deep learning-based studies, our small sample size can be considered another limitation of our study. Future studies could evaluate the performance of the models using larger sample sizes.
Thank you very much for your contribution.
Comment 6.
Conclusion
Provide explicit guidance on the potential use of these AI models in clinical practice, specifying limitations.
Response 6:
Your contribution played an important role in improving our manuscript. Thank you very much.
Based on your suggestion, we have added the following paragraph to the discussion section of our manuscript.
Our study demonstrates that LLMs, such as ChatGPT and Claude, still face significant limitations in their application to medical image analysis. It was observed that neither model, particularly ChatGPT, achieved sufficient diagnostic sensitivity and accuracy in detecting AIS. ChatGPT's notably high false-positive results in the healthy control group and its predictions resembling random guesses pose a risk of unnecessary additional testing and treatment, raising significant clinical concerns.
Nevertheless, our study highlights the potential of these models in detecting AIS from DW-MRI images, contributing significantly to the theoretical and practical knowledge base in the fields of AI and neuroradiology. Despite their current inability to achieve the desired success in medical image analysis, we anticipate that technological advancements will enable these models to reach high success rates in the coming years, playing a crucial role in optimizing the increasing workload of radiologists. In healthcare centers with insufficient radiologist staffing, healthcare personnel with limited clinical experience could utilize these LLMs for the preliminary evaluation of DW-MRI images, thereby expediting diagnostic processes. This innovative technology could also serve as a secondary control mechanism for junior radiologists during their professional development, minimizing the risk of overlooking critical findings and enhancing diagnostic reliability. Additionally, in studies implementing optimized protocols aimed at reducing door-to-puncture time (DTPT) in conditions requiring early intervention, such as AIS (8), AI models like LLMs could be incorporated into the image evaluation process by relevant specialists. However, it must be acknowledged that these models are still in their early stages and are prone to errors.
Comment 7.
Figures Improve the resolution of figures to enhance readability.
Response 7:
Thank you very much for your contribution to improving the quality of the figures.
Based on your suggestion, the resolution of the figures has been enhanced. The updated figures are as follows.

Figure 1. Patient selection process

Figure 2. The process of presenting and evaluating combined DWI and ADC images in a single image with a three-question prompt to the models.
We are grateful for the time you have dedicated to our study. Your recommendations have guided us in preparing a more enhanced and structured version of our study. We hope that our revisions meet your expectations.
Sincerely

Round 2
Reviewer 1 Report
Comments and Suggestions for Authors
Most of my previous comments were answered. However, since the authors are unable to perform evaluations on medical LVLMs, I have another concerns. In the responses, the authors claim that the main purpose of the study is to evaluate the potential and limitations of two 'general-purpose' LVLMs. I would like to request for clarification on the value of this work.
1. Why is it important to evaluate the performance of general LVLMs on medical tasks, given they can be specially fine-tuned for medical tasks?
2.Why two models are enough for the study, given there are many public-available LVLMs other than ChatGPT4o and Claude? Does your ethical committee approval prevent you from testing other public models?
Author Response
Thank you, dear reviewer, for your valuable feedback on our study. We have carefully evaluated your comments and present our detailed responses below.
Comments 1.
1. Why is it important to evaluate the performance of general LVLMs on medical tasks, given they can be specially fine-tuned for medical tasks?
Response 1.
Evaluating the performance of general-purpose LVLMs on medical tasks is necessary for several important reasons:
1. General-purpose LVLMs like ChatGPT and Claude are widely available and can be used by healthcare professionals, researchers, and the general public without the need for advanced technical skills or specialized infrastructure.
2. Assessing the capabilities of these models in medical tasks, such as analyzing medical images, is essential to understanding their potential applications in clinical environments. These evaluations highlight their accuracy, common error tendencies, and limitations in handling complex medical situations.
3. By systematically examining the weaknesses and error patterns of general-purpose models in medical contexts, developers gain valuable insights. This feedback plays a key role in refining the models and optimizing them for specific medical use cases.
Comments 2.
2. Why two models are enough for the study, given there are many public-available LVLMs other than ChatGPT4o and Claude? Does your ethical committee approval prevent you from testing other public models?
Response 2.
Dear reviewer, not evaluating more models is a limitation of our study. We acknowledge this limitation and accept that comparing the performance of additional models could provide a broader perspective. The reason for using these two models in our study stems from their ability to provide solutions to many problems and their considerable popularity in the public sphere. Our ethics committee approval does not explicitly prevent testing other publicly available models. However, our methodology in the ethics committee application specifically aimed to test ChatGPT and Claude's AIS detection ability, and evaluating our dataset with other AI types would not align with this approved methodology. Additionally, in our study, we evaluated the performance of the two models with a two-week interval. Including other models in the study would have significantly extended the completion time. Therefore, additional models could not be included in the scope of the study. This study serves as a foundation for more comprehensive research. The methodology we developed and our findings can be used by other researchers to evaluate additional models and conduct broader studies.
Reviewer 2 Report
Comments and Suggestions for Authors
Authors answered all my comments
Comments on the Quality of English LanguageMinor English check
Author Response
Thank you very much for your contribution.
Reviewer 3 Report
Comments and Suggestions for Authors
Thank you for addressing the concerns raised during the previous review cycle. After carefully evaluating the revised manuscript, I am pleased to acknowledge that the new results and refined analyses have significantly enhanced the overall value and impact of the study.
Author Response

(The authors gave the same response as above.)
